# Controlled Cationic Polymerization of *p*-Methylstyrene in Ionic Liquid and Its Mechanism

**DOI:** 10.3390/polym14153165

**Published:** 2022-08-03

**Authors:** Xiaoqian Zhang, Shengde Tang, Ming Gao, Chunfeng Sun, Jiasheng Wang

**Affiliations:** 1Hebei Key Laboratory of Hazardous Chemicals Safety and Control Technology, School of Chemical Safety, North China Institute of Science and Technology, Yanjiao, Beijing 101601, China; gmscy@hotmail.com (M.G.); sunchf@ncist.edu.cn (C.S.); bluceking@163.com (J.W.); 2China National Pulp and Paper Research Institute Co., Ltd., Beijing 100102, China

**Keywords:** cationic polymerization, *p*-methylstyrene, ionic liquid, controlled polymerization, mechanism

## Abstract

Ionic liquid (IL) as a green solvent is entirely composed of ions; thus, it may be more than a simple solvent for ionic polymerization. Here, the cationic polymerization of *p*-methylstyrene (*p*-MeSt) initiated by 1-chloro-1-(4-methylphenyl)-ethane (*p*-MeStCl)/tin tetrachloride (SnCl_4_) was systematically studied in 1-butyl-3-methylimidazolium bis(trifluoromethanesulfonyl)imide ([Bmim][NTf_2_]) IL at −25 °C. The results show that IL did not participate in cationic polymerization, but its ionic environment and high polarity were favorable for the polarization of initiator and monomer and facilitate the controllability. The gel permeation chromatography (GPC) trace of the poly(*p*-methylstyrene) (poly(*p*-MeSt)) changes from bimodal in dichloromethane (CH_2_Cl_2_) to unimodal in IL, and polydispersities *M*_w_/*M*_n_ of the polymer in IL showed narrower (1.40–1.59). The reaction rate and heat release rate were milder in IL. The effects of the initiating system, Lewis acid concentration, and 2,6-di-*tert*-butylpyridine (DTBP) concentration on the polymerization were investigated. The controlled cationic polymerization initiated by *p*-MeStCl/SnCl_4_ was obtained. The polymerization mechanism of *p*-MeSt in [Bmim][NTf_2_] was also proposed.

## 1. Introduction

Cationic polymerization belongs to chain polymerization, which is an important branch of polymerization reactions. The main solvents for cationic polymerization are chlorinated alkanes, which are corrosive, toxic, and volatile and pollute the environment. Moreover, traditional organic solvent cationic polymerization has poor controllability, therefore limiting its macromolecular design and industrial applications. Thus, the change from a traditional solvent to less conventional medium, such as ionic liquid (IL), deserves to be researched carefully. In recent year, ILs have been recognized as environmentally friendly “green solvents” and have attracted more attention due to their outstanding properties, such as non-volatility, good thermal and chemical stability, negligible flammability, and tunability [1,2,3,4,5,6,7,8,9,10,11]. In cationic polymerization, the distance and interactions between the active center and counter ion are very important for the reaction [12,13,14,15]. IL is entirely composed of ions, and it is regarded as a polar, but non-coordinating, solvent with a high charge density [16,17]. The ionic environment of IL is different from that of traditional molecular solvents, which make IL more than a simple solvent for ionic polymerization. Therefore, the research on cationic polymerization in IL medium is of great significance. 

At present, ILs have been used in various polymerization reactions, such as radical polymerization [18,19,20,21], polycondensation [22,23,24,25], ring opening polymerization, and electrochemical reaction, where many interesting phenomena different from traditional molecular solvents have been found. Some studies show IL cannot only serve as a solvent, but also play a role of catalyst [26,27,28,29]. However, compared with other reactions, IL as a solvent for ionic polymerization is rarely reported. In anionic polymerization, there are few reports mainly because the imidazolium ring of imidazolium ILs is acidic in nature and frequently incompatible with bases, and easily reacts with initiators, resulting in low yield and wide molecular weight distribution [30,31]. Regarding cationic polymerization there have been some reports. For example, Vijayaraghavan first reported the living cationic polymerization of styrene in IL at 0 or 60 °C (polydispersities *M*_w_/*M*_n_~1.3), but *M*_n_ was small (<2500) [32,33]. The report about styrene cationic polymerization in IL revealed that the high polarity of ILs could promote initiator dissociation, but the reaction was not controllable [34]. Recently, the reports on the cationic polymerization of vinyl ethers or *p*-methylstyrene (*p*-MeSt) in ILs showed that IL interacted with the carbocation center, but the reaction was still poorly controlled [35,36,37]. In sum, although some progress about IL as a cationic polymerization solvent has been made, the controllability of the reaction needs to be improved and the mechanism need to be further studied.

*p*-MeSt, as a well-known, commercially available styrene derivative, could undergo cationic polymerization. In this study, the cationic polymerization of *p*-MeSt initiated by the 1-chloro-1-(4-methylphenyl)-ethane (*p*-MeStCl)/tin tetrachloride (SnCl_4_) system was systematically studied in 1-butyl-3-methylimidazolium bis(trifluoromethanesulfonyl)imide ([Bmim][NTf_2_]) ionic liquid. The effects of the initiating systems, Lewis acid concentration, and 2,6-di-*tert*-butylpyridine (DTBP) concentration on the polymerization are investigated. The results show that the gel permeation chromatography (GPC) trace of the product changes from bimodal in dichloromethane (CH_2_Cl_2_) to unimodal in IL, and *M*_w_/*M*_n_ in IL is narrower (1.40–1.59). The polymerization in IL was achieved in a milder way than that in CH_2_Cl_2_, including mild reaction rate and heat release rate. The ionic environment and high polarity of IL facilitate the controllability. The controlled cationic polymerization of *p*-MeSt initiated by *p*-MeStCl/SnCl_4_ in [Bmim][NTf_2_] was obtained. Finally, a possible polymerization mechanism of *p*-MeSt in [Bmim][NTf_2_] was proposed.

## 2. Experimental Section

### 2.1. Materials

[Bmim][NTf_2_], 1-octyl-3-methylimidazolium bis(trifluoromethanesulfonyl)imide ([Omim][NTf_2_]), 1-butyl-3-methylimidazolium tetrafluoroborate ([Bmim][BF_4_]), 1-octyl-3-methylimidazolium tetrafluoroborate ([Omim][BF_4_]), 1-butyl-3-methylimidazolium hexafluorophosphate ([Bmim][PF_6_]), 1-octyl-3-methylimidazolium hexafluorophosphate ([Omim][PF_6_]) and 1-ethyl-3-methylimidazolium trifluoromethanesulfona ([Emim][TA]) (Sigma Aldrich, Beijing, China, >99.0%) all were dried and degassed at 70 °C for a few days on vacuum line until the H_2_O concentration was less than 30 ppm before use [29]. *p*-MeSt (TCI, Shanghai, China, >98.0%) was dried overnight over CaH_2_ (calcium hydride, Beijing Chemical Co., Beijing, China,) and distilled under reduced pressure before use. CH_2_Cl_2_ (Beijing Chemical Co., Beijing, China, >99.9%) and DTBP (Aldrich, Beijing, China, >97.0%) were distilled twice over CaH_2_ under reduced pressure before use. TiCl_4_ (Aldrich, Beijing, China, >99.9%), SnCl_4_ (Aldrich, Beijing, China, >99.0%), and anhydrous methanol (Beijing chemical Co., Beijing, China, >99.9%) were used as received. a,a-Dimethylbenzyl chloride (CumCl, J&K SCIENTIFIC LTD., Beijing, China, >97.0%) was also used with further purification by double distillation from CaH_2_ under reduced pressure. The 2-chloro-2,4,4-trimethylpentane (TMPCl) [38] and *p*-MeStCl [39] were synthesized. The clean and quantitative formation of TMPCl and *p*-MeStCl were confirmed by ^1^H NMR (nuclear magnetic resonance) spectroscopy.

### 2.2. Polymerization

Polymerization experiments were carried out under a dry nitrogen atmosphere ([H_2_O] < 0.5 ppm; [O_2_] < 10 ppm) in an MBraun 150-M glovebox using 50 mL test tubes with an IKA-MS3 vortex stirrer. A typical *p*-MeSt polymerization was carried out in [Bmim][NTF_2_] at −25 °C using the following concentration: [*p*-MeSt]_0_ = 1.880 mol·L^−1^, [*p*-MeStCl]_0_ = 7.427 mmol·L^−1^, [SnCl_4_]_0_ = 133.686 mmol·L^−1^, [DTBP]_0_ = 14.854 mmol·L^−1^. [Bmim][NTF_2_] (4.8 mL), *p*-MeSt (1.6 mL), DTBP stock solution in CH_2_Cl_2_ (0.6 mL; 0.4 mol/L), and *p*-MeStCl stock solution in CH_2_Cl_2_ (0.245 mL; 0.2 mol/L) were added and mixed thoroughly at −25 °C. The polymerization was started by the addition of SnCl_4_ (0.114 mL). After a pre-determined time, the polymerization was terminated with prechilled methanol. The products were washed several times with deionized water, handled under the decompress filter, and then dried in vacuum oven to a constant weight in a few days. Monomer conversions were determined by gravimetric analysis. The details are as follows: monomer conversion (*CVR*_mon_) was calculated according to the constant weight (*m*) of the product, the moles of initiator (*n**_i_*), and the mass (*m*_0_) of the monomer initially added according to:(1)$$CVR_{\mathrm{mon}}=\frac{m-n_{i}\times 150.19}{m_{0}}$$
where mass *m* was determined on a CPA324S balance (Sartorius, Gottingen, Germany) with a precision of ±0.0001 g, and 150.19 corresponds to the total molecular weight of the end structures of product, including -CH(C_6_H_6_)-CH_3_ at the α-end and the -OCH_3_ at the ω-end.

### 2.3. Measurements

Molecular weight and polydispersities (*M*_w_/*M*_n_) were measured by gel permeation chromatography (GPC, Taunton, Massachusetts, USA) equipped with four Waters styragel (500, 10^3^, 10^4^, and 10^5^) in tetrahydrofuran (THF) as an eluent at a flow rate of 1.0 mL/min. The GPC system consists of a Waters e2695 Separations Module, a Waters 2414 RI Detector, and a Waters 2489 UV Detector. Polystyrene standards were used for calibration. FT-IR spectra were recorded by a Nicolet-Nexus 670 spectrometer. The reaction system temperature was detected by the testo 176T4, a data logger: temperature, recording a data for 5 s. ^1^H NMR spectrum was performed on a Bruker AV600 MHz spectrometer (Billerica, Massachusetts, USA) in CDCl_3_ (deuterochloroform) at 25 °C. MALDI-TOF-MS (Matrix-assisted laser desorption ionization time-of-flight mass spectrometry) analysis was carried out on a Ultraflex (AB Sciex, Boston, Massachusetts, USA) time-of-flight mass spectrometer, using 2,5-dihydroxybenzoic acid (DHB) as a matrix, and nitrogen laser desorption at 337 nm. The spectra were recorded in the absence of any cationizing agent. The mass spectra represent averages over 250 consecutive laser shots. External calibrations were performed with peptide calibration standard. The viscosity of ILs was measured by using a Brookfield viscometer (Model DV-II+, Stoughton, MA, USA) with a cone spindle. The mass fraction solubility of *p*-MeSt in ILs was measured using the cloud-point method. The cloud point of solutions was measured by using a polarized optical microscope (Olympus BHSP, Tokyo, Japan) equipped with a temperature controlling stage (Linkam THMS 600, London, UK) under a parallel polarizer.

### 2.4. Computational Method

All calculations were performed with the Gaussian 03 software. Geometry optimizations and vibrational frequency calculations were performed by the density functional theory (DFT) using the B3LYP/3-21G basis set. This provided the charges of carbocation and the atoms’ distances.

## 3. Results and Discussion

### 3.1. Screening of the IL Solvent Suitable for Cationic Polymerization

Generally, the nature of the reaction solvent, especially solubility and viscosity, has an important effect on the reaction. The ability to dissolve monomers directly affects whether the solvent can be used as a polymerization reaction solvent. The viscosity of the medium affects the operation and post-treatment of the reaction, the exothermic conditions in the reaction process, the initiation and termination of the polymerization reaction, etc. Since there are many kinds of ILs, the ILs suitable for cationic polymerization were screened first. In general, cationic polymerization is usually conducted at low temperatures. Therefore, eight typical and commonly used imidazoles ILs with low melting points were selected for further screening. The viscosity of ILs, viscosity of 25% (*v*/*v*) *p*-MeSt in ILs, and monomer solubility were studied, as shown in Table 1. The results show that [Bmim][NTf_2_] has a lower viscosity and a higher ability to dissolve monomers, which is more suitable for cationic polymerization at low temperatures. Therefore, [Bmim] [NTf_2_] was selected for further research.

### 3.2. Effects of the p-MeStCl/SnCl_4_ Initiating System

For the two-component initiating systems, the protonic acid serves as the initiator, and the Lewis acid acts as the activator or catalyst that assists carbocation formation from the initiator and triggers propagation [40,41,42]. The initiating systems is the key to living/controlled cationic polymerization. In this paper, the commonly used initiating system, *p*-MeStCl/SnCl_4_, was studied in IL [Bmim][NTf_2_] and traditional molecular solvent CH_2_Cl_2_. The results are shown in Table 2. We can observe that the polymer prepared in [Bmim][NTf_2_] is obviously smaller in molecular mass distribution (*M*_w_/*M*_n_) and has slightly higher yield compared to traditional solvent CH_2_Cl_2_. It is worth noting that the GPC trace of the polymer using the *p*-MeStCl/SnCl_4_ initiating system in [Bmim][NTf_2_] was unimodal, while that in CH_2_Cl_2_ was bimodal, as shown in Figure 1. The unimodal shape of the GPC curve and low *M*_w_/*M*_n_ are the results of the greater homogeneity of the polymer, which is related to the presence of one type of active center. In the present study, the product obtained in CH_2_Cl_2_ was characterized by ^1^H-NMR (Figure 2). Here, we observed a significant signal at 5.9–6.1 ppm (ω′) assigned to the ethylenic structure (structure II in Figure 2), which was the result of the serious *β*-H elimination from –CH_2_– in the growing carbocation [43,44,45]. The *β*-H produced by side reaction further initiated monomer polymerization, resulting in the bimodal shape of the GPC curve. However, IL entirely consisted of ions and its ion environment influenced the ion pairs of polymerization; IL can stabilize the growing carbocation through the moderation or delocalization of positive charges, which reduces the occurrence of *β*-H elimination. These effects of IL on cationic polymerization were confirmed by theoretical calculations. Thus, we concluded that, compared to the molecular solvent, IL is more conducive to the controllability of the reaction.

### 3.3. The Specialties of IL: Ionic Nature and High Polarity

The active center of the cationic (anionic) polymerization reaction is the cation (anion). Ionic liquid is an ionic compound composed of an anion and cation. Because of its ionic nature, IL is regarded as a polar, but non-coordinating, solvent with a high charge density; thus, IL may be more than a simple solvent used for ionic polymerization. Thus, whether IL was involved in the reaction was analyzed first. When the monomer was introduced into [Bmim][NTf_2_] IL, stirred well, and left for 24 h, the solution remained clear and transparent. This shows that IL itself could not initiate the cationic polymerization as a protonic acid or Lewis acid, even though it is an ion pair. Namely, the [Bmim][NTf_2_] IL was not directly involved in the cationic polymerization and it acted as a solvent.

In order to reveal the differences between IL with an ionic nature and traditional solvents with a molecular environment, we provided a theoretical basis for the effect of IL and CH_2_Cl_2_ on cationic polymerization by the density functional theory (DFT) using Gauss software. Here, the interaction between the ion pair, namely, the growing carbocation *p*-Mest^+^ and its counter ion SnCl_5_^−^, and IL or the CH_2_Cl_2_ solvent was performed. The results are presented in Figure 3. In terms of the active center charge, the charge of the carbocation in IL (+0.198 e) was lower than that in CH_2_Cl_2_ (+0.207 e); in terms of the distance between the ion pairs expressed as the distance between C^+^ in *p*-Mest^+^ and Sn in SnCl_5_^−^, the distance in IL (4.147 Å) was greater than that in CH_2_Cl_2_ (3.976 Å). The results show that IL is different from traditional molecular solvents and its ion environment affects *p*-Mest cationic polymerization. The IL ions interact with the growing carbocation, which affects the distance between ion pairs; the weakly nucleophilic anion of IL can stabilize the rising end of carbocation species through the moderation or delocalization of positive charges, which further explains the formation of a single peak and leads to a low polymerization rate.

Moreover, solvent polarity has an important effect on the rate, activity characteristics, and controllability of cationic polymerization [46]. With the increase in solvent polarity, the reaction rate increases. High solvent polarity is conducive to the polarization of the initiator and monomer, thus accelerating initiation and aggregation. Here, the normalized solvent polarity scale $E_{N\;T}$ with an ordering of [Bmim][NTf_2_] IL and eighteen molecular solvents was summarized (Figure 4) [47,48,49,50]. It can be observed that the polarity of [Bmim][NTf_2_] was obviously higher than that of the traditional solvent CH_2_Cl_2_. Thus, when [Bmim][NTf_2_] was used as a reaction medium, its ionic environment and high polarity were possibly suitable for cationic polymerization, which made it different from traditional molecular solvents. This was maybe the reason why the GPC curve of the polymer initiated by *p*-MeStCl/SnCl_4_ in [Bmim][NTf_2_] was unimodal and the distribution was narrow.

### 3.4. Mild Reaction Rate and Heat-Release Rate

The reaction rate and heat release with reaction time for *p*-MeSt polymerization with the *p*-MeStCl/SnCl_4_ initiating system in [Bmim][NTf_2_] and CH_2_Cl_2_ were studied (Figure 5). Figure 5a shows that, in CH_2_Cl_2_, the reaction was instantaneously completed, while in [Bmim][NTf_2_], the reaction rate was much slower and the conversion rate was slightly higher. Figure 5b shows that the heat-release rate of the polymerization system in [Bmim][NTf_2_] was slower than that in CH_2_Cl_2_. The peak of the system’s temperature in CH_2_Cl_2_ was 18.5 °C, while in [Bmim][NTf_2_], it was lower, only 9.6 °C. The results reveal that the polymerization in ILs is proceeded in a milder way than that in CH_2_Cl_2_. This was maybe because ILs provide an ionic environment, which could stabilize the growing carbocation through the moderation or delocalization of the positive charges. Furthermore, the high viscosity of ILs may have contributed to this difference. This further indicated that ILs are more suitable for cationic polymerization. 

### 3.5. Effects of Lewis Acid Concentration on p-MeSt Polymerization in ILs

The concentration of SnCl_4_ as Lewis acid is one of the most important variables in controlling the polymerization degree, molecular weight and polydispersity. This was investigated and its results, including GPC traces of poly(*p*-methylstyrene) (poly(*p*-MeSt)), are presented in Figure 6. From 0.015 to 0.133 mol·L^−1^, the GPC curves are unimodal. Moreover, at the same polymerization time, both the monomer conversion and *M*_n_ increase, while *M*_w_/*M*_n_ has no obvious change. This suggests that the high concentration of SnCl_4_ is expected to facilitate the activation of the carbocation terminal (---C-Cl→---C···SnCl_5_), and thereby accelerate the polymerization process.

Conditions: [*p*-MeSt]_0_ = 1.88 mol·L^−1^; [*p*-MeStCl]_0_ = 7.427 mmol·L^−1^; T = −25 °C. The solvent was [Bmim][NTF_2_] and all the reaction times were 2 h.

### 3.6. Effects of DTBP Concentration on p-MeSt Polymerization in ILs

2,6-di-*tert*-butylpyridine (DTBP) is a proton scavenger and has been used for controlled/living cationic polymerization to prevent uncontrolled initiations [51,52]. Thus, to achieve controlled *p*-MeSt polymerization in ILs, DTBP was added and the effects of DTBP concentration on *p*-MeSt polymerization with *p*-MeStCl/SnCl_4_ were investigated (Table 3). The results show that the addition of DTBP remarkably increases the molecular weight and narrows the molecular weight distribution. In the absence of DTBP, a 93.1% conversion was reached and the molecular weight was lower than the expected theoretical molecular weight (*M*_n_ = 11,950, *M*_w_/*M*_n_ = 1.52 vs. theoretical *M*_n_, *M*_n,thero_ = 27,930). At [DTBP]_0_ = 0.0149 mol·L^−1^, the molecular weight was close to the theoretical *M*_n_ and a relatively narrow molecular weight distribution was observed. As [DTBP] was higher than 0.0149 mol·L^−1^, the conversion decreased and the *M*_n_ was larger than the theoretical value. This was probably because, after DTBP trapped the proton, complexes (DTBP-H^+^ SnCl_5_^−^) between Lewis acid and DTBP were formed [39,53]. In other words, DTBP conferred “livingness” through the formation of common anions that suppressed propagation via free ions in a fashion similar to *n*-Bu_4_NCl. Thus, in the following research, [DTBP]_0_ = 0.0149 mol·L^−1^ was used.

### 3.7. The Controllability of p-MeSt Cationic Polymerization in ILs

In a series of experiments, the controllability of *p*-MeSt cationic polymerization initiated by the *p*-MeStCl/SnCl_4_ system with the addition of DTBP in IL was studied (Figure 7). The polymerizations were quantitative at their respective initiator concentrations, and the rate of the reaction increased as the *p*-MeStCl concentration increased (Figure 7A). As shown in Figure 7B, the first-order-kinetic plots for *p*-MeStCl/SnCl_4_ initiating system were all linear; it meant that the irreversible termination was absent during the polymerization. Figure 7C showed that the molecular weight of the product poly(*p*-MeSt) increased with monomer conversion, irrespective of the *p*-MeStCl concentration. Moreover, *M*_n_s was inversely proportional to the initiator concentration and nearly approached the theoretically calculated values. In addition, the *M*_w_/*M*_n_ was also independent of the *p*-MeStCl concentration and leveled at the values of 1.35 ≤ *M*_w_/*M*_n_ ≤ 1.59 (Figure 7C).

In order to further investigate the controllability of the reaction, the effects of the feed ratio of *p*-MeSt ([M]) to *p*-MeStCl ([I]) on *M*_n_ and *M*_w_/*M*_n_ were studied (Figure 8). It shows that *M*_n_ was linearly related to the [M]/[I] ratio; the *M*_w_/*M*_n_ remained narrow and leveled in the range of 1.40–1.59; *M*_n_s approached the theoretical values, assuming that one *p*-MeStCl generated a polymer chain and the polymer *M*_n_s were controlled. Moreover, the separate chain extension experiment using the so-called monomer addition experiment was also examined to verify the living nature of the *p*-MeSt cationic polymerization in [Bmim][NTF_2_] (Figure 9). A fresh monomer was added to the reaction mixture before the initial monomer had been completely polymerized. The polymerization proceeded smoothly and finally *M*_n_ doubled, while the *M*_w_/*M*_n_ remained narrow. The above results confirm that the *p*-MeSt cationic polymerization initiated by the *p*-MeStCl/SnCl_4_/DTBP system was well controlled in [Bmim][NTF_2_]. 

Conditions: [*p*-MeStCl]_0_ = 31.830 mmol·L^−1^; [SnCl_4_]_0_ = 133.686 mmol·L^−1^; [DTBP]_0_ = 14.854 mmol·L^−1^; [M]_0_ = [M]_add_ = 1.880 mol·L^−1^. Polymerization time: 120 min and 240 min.

The terminal structure of the polymer obtained using *p*-MeStCl/SnCl_4_/DTBP by quenching the polymerization with excess methanol in IL was examined via ^1^H NMR spectrum and MALDI-TOF-MS (Figure 10). The ^1^H NMR spectrum (Figure 10A) showed the characteristic of poly(*p*-MeSt) with signals ascribed to main-chain aliphatic protons (b and c), phenyl groups (d), and methyl group -CH_3_ protons attached to the aromatic ring (e). In addition, small signals resulting from the end groups were visible: CH_3_- (*α*; 1.0 ppm) at the *α*-end because of the initiator (*p*-MeStCl), methoxy group -OCH_3_ (*ω*; 2.9–3.0 ppm) at the *ω*-end resulting from the quenching agent methanol, and the terminal -CH proton (f; 4.1–4.3 ppm) attached to the -OCH_3_ groups [54,55]. MALDI-TOF-MS showed a higher sensitivity for the detection of polymers. Figure 10B shows that the spectrum consisted of a series of sharp peaks each separated by a 118.17 *m*/*z* units interval, which corresponds to the molecular weight of the *p*-MeSt monomer. The molecular weight of each individual peak was very close to the calculated value for CH_3_-CPhCH_3_(p)-(-CH_2_-CHPhCH_3_(p)-)_n_-OCH_3_ + Na^+^, i.e., corresponding to the poly(*p*-MeSt) with the initiator moiety at the *α*-end and the methoxy group at the *ω*-end, along with Na^+^ from the MALDI matrix component [56,57,58]. These results also confirm that *p*-MeSt polymerization with the *p*-MeStCl/SnCl_4_/DTBP system proceeded in a controlled fashion.

### 3.8. Mechanism of p-MeSt Cationic Polymerization in ILs by p-MeStCl/SnCl_4_

Taking the obtained results into account, we proposed the corresponding mechanism of *p*-MeSt cationic polymerization initiated by the *p*-MeStCl/SnCl_4_ system in [Bmim][NTf_2_] (Figure 1). Thus, the polymerization started with the interaction between the initiator (*p*-MeStCl) and the metal halide (SnCl_4_), by which SnCl_4_ “activates” the C-Cl bond of *p*-MeStCl generating a dissociated species containing carbocation and metal halide-based counterion. Then, the formed dissociated species reacted with the monomer *p*-MeSt through its carbocation center and commenced the propagation. When DTBP was introduced, the polymerization was controlled. Throughout the reaction, the cation of IL could not trigger a reaction; the anion of IL, a very weakly nucleophilic species, did not participate in the termination reaction. Even so, comparing with traditional molecular solvent CH_2_Cl_2_, IL was favorable to the polymerization (GPC trace in IL was unimodal, while in CH_2_Cl_2_ it was bimodal; the polymer has an obviously narrow polydispersity in IL). This was because there may be an interaction between growing carbocation with IL and the weakly nucleophilic anion of IL, which could stabilize the growing end of the carbocationic species.

## 4. Conclusions

The cationic polymerization of *p*-MeSt initiated by *p*-MeStCl/SnCl_4_ in [Bmim][NTf_2_] ionic liquid was systematically investigated. The results show that the ionic liquid did not participate in the cationic polymerization, but its ionic environment and high polarity were favorable for the polarization of initiator and monomer and facilitate the controllability. The GPC trace of the poly(*p*-MeSt) changes from bimodal in CH_2_Cl_2_ to unimodal in ILs, and the *M*_w_/*M*_n_ of the polymer in IL showed narrower (1.40–1.59). Compared with the organic solvent, in ionic liquid, the reaction rate and heat release rate were milder. Through the study on the effects of following factors, the initiating system, Lewis acid concentration, and DTBP concentration on the polymerization, the controlled cationic polymerization initiated by *p*-MeStCl/SnCl_4_ was obtained. The living nature of the polymerization in ionic liquid was confirmed by the successful monomer addition experiment as well as from the study of the effects of the feed ratio of monomer to initiator on obtained poly(*p*-MeSt). Finally, the possible polymerization mechanism of *p*-MeSt in [Bmim][NTf_2_] was also proposed. Ionic liquids show great advantages and potential for controlled cationic polymerization.

## Data Availability

The data presented in this study are available on request from the corresponding author.

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
