# Peer review of "Controlled Cationic Polymerization of *p*-Methylstyrene in Ionic Liquid and Its Mechanism"

_polymers, 2022, doi:10.3390/polym14153165_

Round 1
Reviewer 1 Report
This paper reports the controlled/living cationic polymerization of p-methylstyrene in ionic liquid with p-MeStCl/SnCl4 initiating system. By optimizing the polymerization condition, the authors succeeded in obtaining poly(p-MeSt) with relatively narrow dispersity. The chain extension experiment confirmed the living nature of the polymerization system, and the MALDI-TOF MS analysis the perfect end group fidelity. Further, by comparing with the similar polymerization system in CH2Cl2, the authors successfully demonstrated the advantage of using IL as the solvent. Overall, the experiments seem to be successfully done, and the results described in the paper are reasonable. The reviewer thinks that this manuscript is worth publishing in Polymers. Below are few comments that the authors need to consider before publication.
1. Section 3.2 discusses the differences in polymerization properties in IL and dichloromethane. However, this argument is superficial and insufficient. The authors should characterize the product obtained in dichloromethane in detail to disclose why this polymerization was unsuccessful, and the results should give hints on why IL is an excellent solvent for cationic polymerization.
2. It would be nice if the authors comment on the effect of the reaction temperature on the polymerization properties. In this paper, the authors only examined polymerization at -25 degC. In general, cationic polymerization need to be performed in low temperature. So, if the controlled/living nature in IL could be maintained even at room temperature, it would be another highlight of this paper.
Reviewer 2 Report
The research concerns the use of imidazolium liquid in the cationic polymerization of p-methylstyrene (p-MeSt), the influence of initiating systems, Lewis acid concentration, and the concentration of 2,6-di-tert-butylpyridine (DTBP) on polymerization. It has been shown that the use of ionic liquid enables the reaction to be carried out under milder conditions, compared to the use of CH2Cl2, and leads to obtaining a monomodal polymer with a narrow Mw/Mn. The mechanism of the p-MeSt polymerization reaction in the ionic liquid is also proposed.
In recent years, the popularity of ionic liquids as the so-called green solvents dropped significantly, which is undoubtedly associated with the costs of their production and disposal. However, they can still be viewed as alternative solvents to carcinogenic alkyl chlorides such as CH2Cl2.
My comments and suggestions:
Line 31. The Authors wrote: "In the last year ...", however, the cited literature covers the years 2004-2017. It is worth supplementing this fragment with newer literature data.
Line 46-47. The authors argue that “… the proton at position 2 of the imidazole ring of the imidazolium ILs is acidic…“. It would be more appropriate to use the term acidic in nature.
Why did the Authors choose [Bmim] [NTf2]? It requires a broader justification.
Lines 61-69. The introduction contains the results of the research that should be included in the conclusions.
Line 96. The Authors wrote that "The monomer conversions were determined by gravimetric analysis." Please describe it in a more detailed way.
Please explain, was the reaction with the ionic liquid carried out without CH2Cl2 or was it added in small amounts?
Lines 122-128. The monomodal shape of the GPC curve and low molecular weight dispersity (Mw/Mn) are the results of greater homogeneity of the polymer, which is related to the presence of one type of active center. Please explain in a more detailed way the role of the ionic liquid in the formation of a homogeneous polymer.
Scheme 1. Is methonal should be methanol.
The Authors indicate that the polymerization in CH2Cl2 was completed immediately, while in [Bmim] [NTf2] the reaction was much slower and the degree of conversion was slightly higher. Were only the ionic nature of the liquid and its much higher viscosity than CH2Cl2 the factors determining the better conversion? In order to confirm this thesis, were the studies carried out using other types of ionic liquids?
The Authors indicate that the interaction between the growing carbocation with IL and the weakly nucleophilic anion of IL can stabilize the rising end of carbocation species. This requires a better explanation and confirmation.
In my opinion, the work should be supplemented. The authors should clearly indicate the aspects of novelty and the possibility of using ionic liquids in the cationic polymerization of styrene.
Round 2
Reviewer 1 Report
I have read the responses to my comments and the revised manuscript. The authors have made revisions in responses to the comments raised in my first-round review. The revised manuscript can now be accepted for publication in Polymers.
Reviewer 2 Report
In my opinion, the manuscript has been sufficiently improved to warrant publication in Polymers. The revised manuscript was changed according to my remarks.